# Anaplastic Lymphoma Kinase Inhibitor-Induced Neutropenia: A Systematic Review

**DOI:** 10.3390/cancers15204940

**Published:** 2023-10-11

**Authors:** Fabien Moinard-Butot, Simon Nannini, Cathie Fischbach, Safa Abdallahoui, Martin Demarchi, Thierry Petit, Laura Bender, Roland Schott

**Affiliations:** Department of Medical Oncology, Institut de Cancérologie Strasbourg Europe, 17 Rue Albert Calmette, 67033 Strasbourg, France; f.moinard-butot@icans.eu (F.M.-B.); s.nannini@icans.eu (S.N.); c.fischbach@icans.eu (C.F.); s.abdallahoui@icans.eu (S.A.); m.demarchi@icans.eu (M.D.); t.petit@icans.eu (T.P.); l.bender@icans.eu (L.B.)

**Keywords:** ALK inhibitor, neutropenia, NSCLC, haematological toxicities

## Abstract

**Simple Summary:**

ALK inhibitors have improved survival and quality of life for patients compared to chemotherapy. These treatments can cause haematological toxicities, particularly neutropenia. The management of side effects to avoid treatment interruptions is essential. Our literature review aims to improve the knowledge of ALK inhibitor-induced neutropenia in order to improve their management.

**Abstract:**

Lung cancers with ALK rearrangement represent less than 5% of all lung cancers. ALK inhibitors are currently used to treat first-line metastatic non-small cell lung cancer with ALK rearrangement. Compared to chemotherapy, ALK inhibitors have improved progression-free survival, overall survival, and quality of life for patients. The results of several phase 3 studies with a follow-up of over 6 years suggest that the life expectancy of these patients treated with targeted therapies is significantly higher than 5 years and could approach 10 years. Nevertheless, these treatments induce haematological toxicities, including neutropenia. Few data are available on neutropenia induced by ALK inhibitors and on the pathophysiological mechanism and therapeutic adaptations necessary to continue the treatment. Given the high efficacy of these treatments, managing side effects to avoid treatment interruptions is essential. Here, we have reviewed the data from published clinical studies and case reports to provide an overview of neutropenia induced by ALK inhibitors.

## 1. Introduction

Lung cancer is one of the most frequently diagnosed cancers and the leading cause of cancer-related death worldwide, with a 5-year OS rate of <20% for newly diagnosed patients [1]. Histologically, lung cancer is divided into two subtypes: small cell lung cancer (15–20%) and non-small cell lung cancer (NSCLC) (80–85%). In addition, there has been evidence of oncogenic drivers, such as EGFR mutation (10–15% NSCLC), BRAF mutation (approximately 7% NSCLC), ALK rearrangement (3–8% NSCLC), and ROS1 translocation (approximately 2% NSCLC), that confer sensitivity to tyrosine kinase inhibitors (TKIs) [2]. The anaplastic lymphoma kinase (ALK) gene encodes a receptor tyrosine kinase enzyme that was discovered in 2007 [3]. In NSCLC, alteration of this gene leads to pro-oncogenic features. EMLA is a common gene fusion partner of ALK. Clinical features associated with this distinct subgroup of NSCLC patients include young age, nonsmoking history, and more frequent brain metastasis. In first-line metastatic lung cancer with ALK rearrangement, there is an overall survival benefit from ALK inhibitors versus chemotherapy [4,5,6]. More recently, second- and third-generation ALK inhibitors have shown a survival benefit over first-generation inhibitors. ESMO and NCCN recommend alectinib, brigatinib, or lorlatinib as a first-line treatment for metastatic ALK-rearranged lung cancer. Ceritinib and crizotinib are other treatment options [7,8].

The historical 5-year OS rate for molecularly unselected stage IV NSCLC is approximately 2% [9]. With chemotherapy, the median overall survival is approximately 20 months [10,11]. With second- and third-generation ALK inhibitors, the overall survival of patients with ALK-translocated NSCLC is beyond 55 months [12,13,14]. With a median follow-up time of 68.6 months with alectinib and 68.0 months with crizotinib, the median overall survival has still not been reached [15]. However, these inhibitors can cause side effects such as gastrointestinal and hepatic complications, fatigue, upper respiratory infections, and visual impairment that can lead to treatment interruptions and reduced efficacy [12,16,17,18]. Among them, neutropenia induced by ALK inhibitors has been reported, but its incidence and management are not addressed in recommendations. However, given the effectiveness of these treatments, it is necessary to obtain a better understanding of this side effect to optimize its management and diminish the risk of prolonged interruption.

We performed a systematic review of neutropenia induced by ALK inhibitors to assess the incidence of grade 3–4 and febrile neutropenia and to assess data on the management of this toxicity.

## 2. Materials and Methods

PubMed was systematically searched for articles published up to 19 March 2023. The keywords used for searching were “crizotinib”, “ceritinib”, “alectinib”, “brigatinib”, “lorlatinib”, “entrectinib”, “ensartinib”, “repotrectinib”, “foretinib”, “TQ-B3139”, “ALK inhibitor”, “neutropenia”, and “safety”. We used the Preferred Reporting Items for Systematic Reviews and Meta-Analyses (PRISMA) method. We summarized primary prospective and retrospective articles to provide a systematic overview of the subject. The full search strategy is provided in Figure 1.

## 3. Results

Overall, 613 records were identified from PubMed. The titles and abstracts of 583 records were screened, and 532 were excluded. The full texts of the remaining 97 reports were assessed, and 46 articles did not report neutropenia. Thus, the full texts of 51 reports were assessed, and these studies were considered eligible. An overview of the results is presented in Table 1. In our systematic review, 51 articles on ALK inhibitor-induced neutropenia were reviewed. Their main points are summarized in the following sections in parallel with a description of the case series.

### 3.1. Population Characteristics

A total of 8593 patients were treated with ALK inhibitors in the clinical studies reported in this review. In all, 8398 (98%) were adults and 195 (2%) were children; 7865 patients (92%) were treated for lung cancer, of whom 7270 (92%) had an ALK rearrangement, 290 (4%) had an ROS1 rearrangement and 305 (4%) had another molecular abnormality, such as in RET, MET, or NTRK. In addition, 728 patients (8%) were treated for another type of tumour, such as lymphoma, brain tumour, sarcoma, or kidney cancer. The majority of patients had received prior systemic therapy (5599 patients, 65%), and 2994 patients (35%) had received ALK inhibitor therapy as a first-line therapy.

In total, neutropenia of any grade was reported in 1058 patients (12%). Grade 3 or 4 neutropenia was reported in 557 patients (6%) (Table 1).

### 3.2. First-Generation ALK Inhibitors Induce Neutropenia

Neutropenia of any grade is reported in 3 to 41% of patients treated with the first-generation ALK inhibitor crizotinib. Grade 3/4 neutropenia is reported in 0 to 30% of patients (Table 1). Neutropenia is one of the most common side effects of crizotinib treatment, as shown in the data from a phase 1 study of crizotinib in ALK-positive NSCLC patients. Thirty-six patients experienced grade 3 or 4 treatment-related adverse events (TRAEs), including nine patients with neutropenia (one with grade 4) [19]. Moreover, in all studies comparing ALK inhibitors to chemotherapy, neutropenia remained more common in those treated with chemotherapy. In a phase 3 study, grade 3 neutropenia was reported in 16.3% of patients administered crizotinib and 20.8% of patients administered chemotherapy [30]. Few data have been reported on the occurrence of neutropenia according to the sex of the patient. Blackhall et al. [23] reported a higher frequency of neutropenia (≥10% points higher) in females than in males in the PROFILE 1005 study. Regarding the timing of neutropenia onset, in a retrospective study in Japan, neutropenia developed within 12 weeks in 210 patients, 12–24 weeks in 31 patients, and 24–52 weeks in 36 patients. The discontinuation rate for crizotinib due to neutropenia was 7.9% [38]. Neutropenia induced by ALK inhibitors is sometimes responsible for treatment interruption and may require dose reduction. In a phase 1 study of crizotinib in ALK-positive NSCLC patients, 20% of patients needed a dose reduction because of neutropenia (n = 2) [18]. In the phase 1 study in children with ALK-positive tumours, crizotinib dose reductions due to toxicity were reported for four patients with grade 4 neutropenia. Two patients were administered crizotinib at 365 mg/m^2^, one patient at 280 mg/m^2^, and one patient at 165 mg/m^2^. Neutropenia arrived between cycles 1 and 7 [22]. In the PROFILE 1005 trial, neutropenia (4%) was one of the TRAEs most commonly associated with dose reductions. The most frequent TRAE associated with dosing interruptions was neutropenia (11%) [22]. Dose reductions were performed due to neutropenia (8.8%; n = 3; all grade 3) in the EUCROSS trial [28].

### 3.3. Second- and Third-Generation ALK Inhibitors Induce Neutropenia

Between 0 and 26% of patients treated with second- and third-generation ALK inhibitors have been reported to exhibit neutropenia of any grade, and between 0 and 20% of patients have been reported to have grade 3 and 4 neutropenia (Table 1). Regarding second-generation ALK inhibitors, in a real-world surveillance study of alectinib in Japan, grade 1 neutropenia was reported in 50/1221 patients (4.1%), grade 2 neutropenia events in 41/1221 patients (3.4%), and grade ≥ 3 neutropenia events in 14/1221 patients (1.1%). The median time from the start of treatment to the onset of these events was 12.0 days (range 1–550). Overall, 93.3% of events resulted in patient recovery or improvement, with a median time from onset to outcome of 28.5 days (range 1–617) [47]. In a phase 1/2 trial of alectinib, dose-limiting neutropenia was reported in one patient in the 900 mg twice daily alectinib cohort who developed grade 3 neutropenia. In the 900 mg bridging cohort, one patient developed grade 3 neutropenia that required a delay in administration of more than 7 days [41]. When looking at the data for third-generation ALK inhibitors, all grades of neutropenia have been observed in between 0 and 7% of cases. Grade 3/4 neutropenia has been reported in between 0 and 1% of cases (Table 1).

### 3.4. ALK Inhibitors Induce Neutropenia in Asian Populations

There are differences in the efficacy of ALK inhibitors between Asian and non-Asian populations in clinical studies, as higher objective response rates (ORRs) have been frequently reported in Asian patients than in non-Asian patients [19,30]. Regarding side effects, one study reported the efficacy and safety data from the phase 3 PROFILE 1007 study of second-line treatment and the PROFILE 1014 study of first-line treatment of metastatic NSCLC. Regarding neutropenia, in the PROFILE 1007 study, crizotinib-induced neutropenia of any grade occurred in 27% of patients, while grade 3/4 neutropenia occurred in 13%. In an Asian population, neutropenia of all grades occurred in 38% of patients, and grade 3/4 neutropenia was reported in 19%. This difference was not found in the PROFILE 1014 study, with 21% of patients exhibiting neutropenia of any grade in general and Asian populations [11].

### 3.5. Febrile Neutropenia

Febrile neutropenia has been reported in 0 to 10% of patients treated with first- and second-generation ALK inhibitors (Table 2). No febrile neutropenia has been described with third-generation ALK inhibitors.

### 3.6. Neutropenia Management

There are few data on the management of neutropenia induced by ALK inhibitors. Rindone et al. [32], in a retrospective single-centre study, described grade 1/2 neutropenia in 2/27 patients and grade 3/4 neutropenia in 8/27 patients treated with crizotinib. The authors described neutropenia that resolved spontaneously within a few weeks or required a short course of oral steroids. Neutropenia usually developed within 6 months of starting crizotinib. They also described two cases of late neutropenia after 8 months and 60 months of crizotinib treatment [32]. In another retrospective study, six patients had grade 3 neutropenia, and crizotinib was interrupted until the neutropenia levels decreased to grade 2 and then continued at the same dose without recurrence [58]. A case report described a patient treated with third-line crizotinib (500 mg/day). At 36 weeks, he developed grade 4 neutropenia. After discontinuation of crizotinib, neutrophils normalized, but upon reintroduction at 400 mg/day, the patient developed grade 4 neutropenia. A further reduction in dose to 250 mg/day allowed the patient to continue treatment for 20 weeks before disease progression [59]. Another case report described a patient treated with crizotinib at 500 mg/day who developed neutropenia after 2 weeks of treatment. Prednisolone 10 mg/day was introduced, and then crizotinib was resumed with a partial response at 6 months [60]. No toxicity management data were found for second- and third-generation ALK inhibitors.

### 3.7. Physiopathological Hypotheses

Crizotinib inhibits hepatocyte growth factor (HGF), which plays a role in haematopoiesis, and inhibits mesenchymal epithelial growth factor (cMET), which prevents neutrophil recruitment to the tumour [61]. Neutrophils also play a role in promoting cancer growth by releasing growth factors, including epidermal growth factor, hepatocyte growth factor (HGF), and platelet-derived growth factor [62]. Inflammation is known to promote tumour development and angiogenesis and to inhibit apoptosis [63]. Studies show that high neutrophil counts or a neutrophil-to-lymphocyte ratio ≥ 3 are associated with a poor prognosis [64]. Only one retrospective study of 36 patients showed a prognostic role for neutropenia induced by ALK inhibitors [58]. An immune-related cause has also been described in the literature. Toyota et al. [60] reported on a patient who developed neutropenia after crizotinib treatment. The neutropenia improved only with low-dose steroids [59]. Zeng et al. [65] reported an activated pathway in sepsis involving AKL and STING in the involvement of innate immunity. Using several ALK inhibitors, such as ceritinib and brigatinib, they were able to modulate the activity of STING and thus the innate immune response [65]. Interestingly, one team identified a nonreceptor tyrosine kinase as a potential target to suppress neutrophils. In this study, lorlatinib inactivated FES signalling in neutrophils and suppressed neutrophil expansion in the bone marrow. They demonstrated that lorlatinib reduces tumour-induced granulopoiesis and neutrophil motility [66]. Other ALK inhibitors have shown activity with low concentrations on FES (IC_50_ < 10 nM). FES is a proto-oncogene present in the myeloid lineages of haematopoietic cells and plays a role in the innate immune response and in myeloid differentiation [67]. These off-target effects of ALK inhibitors may explain the reported neutropenia.

## 4. Discussion

ALK inhibitors have revolutionized the management of cancers with ALK rearrangement abnormalities, improving PFS and OS in patients compared with chemotherapy. These treatments are currently recommended as a first-line therapy for ALK lung cancer. The management of even rare adverse events, such as neutropenia, is essential for good adherence to these treatments to improve patient survival.

Neutropenia is more frequent with first-generation ALK TKIs than with second- and third-generation TKIs [12,53]. Crizotinib most likely induces haematological toxicity and notably neutropenia. However, these treatments are still better tolerated than chemotherapy, with significantly less haematological toxicity [5,10]. Furthermore, the incidence of neutropenia induced by second-generation ALK inhibitors like ceritinib, alectinib, and brigatinib or third-generation ALK inhibitors like lorlatinib is minimal. The incidence of febrile neutropenia is low, particularly for second- and third-generation treatments, with less than 10% of cases reported.

Due to its rarity, very few data on the management of neutropenia during treatment with ALK inhibitors are available. Moreover, all data concern crizotinib and not second- or third-generation ALK inhibitors. According to the cases reported, management depends, as usual, on the toxicity grade. For grade 3 and higher, treatment is usually stopped until neutrophil count normalization or at least less than or equal to that of grade 2. With a half-life of 40 h, neutropenia induced by crizotinib usually persists for 3 to 4 weeks after treatment arrest. In cases of prolonged neutropenia, there are reports of a benefit of short-term low-dose steroids. After recovery, one patient with grade 3 neutropenia resumed crizotinib at the same dose without recurrence. Other reported cases have reduced the dose of treatment, but one of them, with grade 4 neutropenia, showed recurrence of neutropenia. The dose dependence of neutropenia and crizotinib is not clear and dose reduction may favour loss of efficacy more than prevention of this toxicity. For second-generation TKIs, data have been reported on the use of ceritinib and alectinib, but there is no information on management or specific recommendations. Brigatinib and third-generation lorlatinib appear to be less likely to cause neutropenia.

Regarding the prognostic value of induced neutropenia, the scarcity of data makes it impossible to conclude any correlation. In other situations, treatment-induced side effects may have predictive value. The example of high blood pressure has been reported as an efficacy marker for bevacizumab or some anti-VEGF TKIs [68,69]. In another example, in metastatic renal cell carcinoma, TKI-induced high blood pressure may be a good predictor for a better prognosis in patients [70]. For ALK inhibitors, only one case report has described an association between the occurrence of neutropenia and improved progression-free survival with crizotinib treatment [58]. For second- and third-generation therapies, despite a clear relationship between treatment concentration and progression-free survival, especially for alectinib and crizotinib, no evidence has been reported on side effects and efficacy [71]. Additionally, risk factors have not been clearly identified. Hypotheses about a sex or ethnic predisposition to neutropenia require further exploration.

Pharmacovigilance studies would remain the best way to assess the haematological toxicity of ALK inhibitors and obtain a better understanding of the underlying management.

In view of the data in the literature, we propose the management of neutropenia induced by ALK inhibitors. In the event of neutropenia below 500 mm^3^, we propose temporary interruption of treatment, followed by resumption with a reduction in dosage. In the event of recurrence, we suggest the introduction of corticosteroid therapy. This management is essential to maintain treatment, given its survival benefit for patients. There are no data in the literature on changing ALK inhibitor treatment. In our experience, in the event of recurrence of neutropenia despite discontinuation of treatment, changing the molecule does not eliminate this haematological toxicity.

## 5. Conclusions

Neutropenia induced by ALK inhibitors is a common side effect of first-generation therapies and less common with second- and third-generation therapies. Dose reduction is sometimes required to continue treatment, and in a few patients, a low dose of corticosteroids allowed treatment to continue. There are fewer data on the new generation of ALK inhibitors currently used in practice, but these treatments seem less likely to cause neutropenia. Given the metastatic survival data on lung cancer treated with ALK inhibitors, it is important to manage the toxicity associated with these treatments. Several pathophysiological hypotheses have been described, but further studies are needed, especially pharmacovigilance studies.

## Figures and Tables

**Figure 1 cancers-15-04940-f001:**
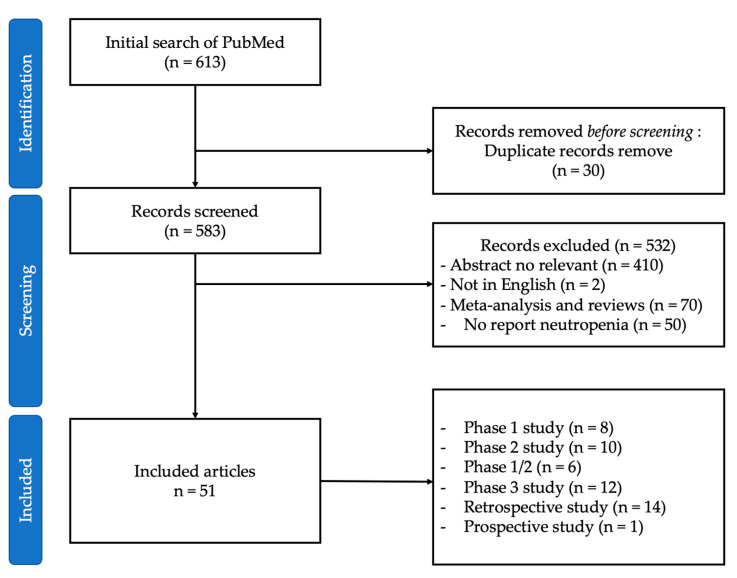
Study selection.

**Table 1 cancers-15-04940-t001:** ALK inhibitors induced neutropenia of all grades and grades 3–4.

Ref.	Study Design	Population Treated with ALK Inhibitors	Cancer Type	Lines of Treatment before Metastatic Disease	Treatment and Dose	Region	Neutropenia All Grade n (%)	Neutropenia G3–4 n (%)
[19]	Phase 1, single-arm, multicentre	Adults n = 149	ALK-positive lung cancer	16% of patients had no previous treatment regimen Other patients had 1–4 or more previous lines of treatment	Crizotinib 250 mg twice per day	USA, Australia, and South Korea	NR	9 (25)
[20]	Phase 1, open-label, expansion cohort, multicentre	Adults n = 53	ROS1-rearranged lung cancer	13% of patients had no previous treatment 65% of patients had 1–2 previous lines of treatment 23% of patients had 3 or more previous lines of treatment	Crizotinib 250 mg twice per day	International	8 (15)	5 (9)
[21]	Phase 1b, open-label, single-arm, multicentre	Adults n = 44	ALK-positive tumours (excluding lung cancer)	16% of patients had no prior systemic therapy Other patients had 1 or more prior lines of treatment	Crizotinib 250 mg twice per day	International	14 (32)	10 (23)
[22]	Phase 1, open-label, multicentre	Children n = 65	ALK-positive solid tumours, lymphomas, CNS tumours	Refractory to therapy	Dose escalationcrizotinib 100, 130, 165, 215, 280, and 365 mg/m^2^ twice per day	USA	21 (32)	10 (15)
[23]	Phase 2, multicentre, single-arm	Adults n = 1069	ALK-positive lung cancer	25% of patients had 1 prior line of systemic therapy 74% had 2 or more prior lines of treatment	Crizotinib 250 mg twice per day	International	226 (21)	137 (13)
[24]	Phase 2, multicentre, nonrandomized, open-label	Adults n = 34	MET alterations clear cell sarcoma	26% had previous systemic therapy	Crizotinib 250 mg twice per day	Europe	6 (18)	2 (6)
[25]	Phase 2, multicentre, single-agent, open-label	Adults n = 48	Rearrangement of TFE3 alveolar soft part sarcoma	48% had received systemic therapy	Crizotinib 250 mg twice per day	Europe	8 (17)	2 (4)
[26]	Phase 2, single-arm, multicentre	Adults n = 127	ROS1-rearranged lung cancer	19% of patients had no prior therapy 42% had 1 prior line of therapy 39% had 2 or more prior lines of therapy	Crizotinib 250 mg twice per day	East Asia	43 (34)	15 (12)
[27]	Phase 2, multicentre, single-arm	Adults n = 20	Inflammatory myofibroblastic tumours	40% of patients had prior systemic therapy	Crizotinib 250 mg twice per day	Europe	4 (20)	2 (10)
[28]	Phase 2, multicentre, single-arm	Adults n = 34	ROS1-rearranged lung cancer	21% of patients had no prior therapy Other patients had 1 or more prior lines of therapy	Crizotinib 250 mg twice per day	Europe	11 (32)	3 (9)
[29]	Phase 2, open-label, randomized	Adults n = 152	Papillary renal cell carcinoma	7% of patients had prior systemic therapy	Crizotinib 250 mg twice per day	USA and Canada	1 (3)	0 (0)
[5]	Phase 3, randomized, open-label, multicentre	Adults n = 171	ALK-positive lung cancer	No prior therapy	Crizotinib 250 mg twice per day	International	42 (25)	26 (15)
[30]	Phase 3, randomized, open-label, multicentre	Adults n = 104	ALK-positive lung cancer	No prior therapy	Crizotinib 250 mg twice per day	East Asia	43 (41)	17 (16)
[31]	Prospective, multicentre study	Adults n = 11	ALK-positive anaplastic large T-cell lymphoma	Previous chemotherapy	Crizotinib 250 mg twice per day	Europe and North America	2 (18)	0 (0)
[32]	Retrospective, monocentric	Adults n = 27	ALK-positive anaplastic large T-cell lymphoma Lymphoma	2 median previous lines (range 1–6)	Crizotinib 250 mg twice per day	Europe	10 (37)	8 (30)
[33]	Retrospective, national cohort	Adults n = 90	c-MET and ROS1-positive lung cancer	Between 1–7 previous lines of systemic therapy	Crizotinib 250 mg twice per day	France	21 (23)	9 (10)
[34]	Retrospective, monocentre	Adults n = 35	ROS1-rearranged lung cancer	49% of patients had 1 previous line of therapy 31% of patients had 2 previous lines of therapy 20% of patients had 3 or more previous lines of therapy	Crizotinib 250 mg twice per day	China	5 (14)	1 (3)
[35]	Retrospective, monocentre	Adults n = 7	ALK-positive lung cancer	No prior therapy	Crizotinib 250 mg twice per day	China	1 (14)	0 (0)
[36]	Chart review, retrospective study	Adults n = 38	ALK-positive lung cancer	55% had prior chemotherapy	Crizotinib 250 mg twice per day	Kuwait and Saudi Arabia	6 (16)	0 (0)
[37]	Retrospective, single-centre	Adults n = 104	ALK-positive lung cancer	61% of patients had no prior therapy Other patients had 1 or more prior lines of therapy	Crizotinib 250 mg twice per day	China	20 (19)	3 (3)
[38]	Retrospective, multicentre	Adults n = 2028	ALK-positive lung cancer	28% of patients had no prior lines of therapy Other patients had 1 or more prior lines of therapy	Crizotinib 250 mg twice per day	Japan	278 (14)	183 (9)
[39]	Retrospective, multicentre	Adults n = 91	ALK-positive lung cancer	44% of patients had no prior chemotherapy	Crizotinib 250 mg twice per day	Spain	8 (9)	4 (4)
[40]	Phase 1/2, multicentre, open-label	Adults n = 28	RET-rearranged lung cancer	21% of patients had 1 prior line of chemotherapy 79% had 2 or more prior lines of chemotherapy	Alectinib 450 mg twice per day	Japan	1 (4)	1 (4)
[41]	Phase 1/2, single-arm, open-label	Adults n = 47	ALK-positive lung cancer	13% of patients had no prior therapy 57% had 1 or 2 prior lines of therapy 30% had 3 or more prior lines of therapy	Dose escalation Alectinib 300 mg, 460 mg, 600 mg, 760 mg, 900 mg twice per day	USA	3 (6)	2 (4)
[42]	Phase 1/2, single-arm, open-label	Adults n = 58	ALK-positive lung cancer	2% of patients had no prior therapy Other patients had 1 or more prior lines of therapy	Dose escalation Alectinib 20 to 300 mg twice daily	Japan	15 (26)	4 (7)
[43]	Phase 2, multicentre	Adults n = 18	ALK-positive lung cancer with poor performance status	72% had no previous systemic therapy 28% had chemotherapy, crizotinib or both	Alectinib 300 mg twice per day	Japan	3 (17)	0 (0)
[44]	Phase 2, open-label, multicentre	Children and Adults n = 10	ALK-positive anaplastic large T-cell lymphoma	1 or 2 previous lines of systemic therapy	Alectinib 600 mg twice per day	Japan	2 (20)	2 (20)
[13]	Phase 2, multicentre, single-agent, open-label	Adults n = 87	ALK-positive lung cancer, crizotinib-resistant	74% of patients had previous chemotherapy	Alectinib 600 mg twice per day	USA and Canada	4 (5)	1 (1)
[4]	Phase 3, multicentre, randomized, open-label	Adults n = 70	ALK-positive lung cancer, crizotinib pretreated	2 prior lines of systemic therapy	Alectinib 600 mg twice per day	Europe and Asia	2 (3)	0 (0)
[45]	Phase 3, multicentre, randomized, open-label	Adults n = 303	ALK-positive lung cancer	No prior therapy	Alectinib 600 mg twice per day or Crizotinib 250 mg twice per day		NR	Alectinib: 0 (0) vs. Crizotinib: 8 (5)
[17]	Phase 3, multicentre, randomized, open-label	Adults n = 207	ALK-positive lung cancer	64% of patients had no prior systemic therapy Other patients had 1 line of systemic therapy	Alectinib 600 mg twice per day or Crizotinib 250 mg twice per day	Japan	Alectinib: 3 (3) vs. Crizotinib: 19 (18)	Alectinib: 2 (2) vs. Crizotinib: 14 (14)
[46]	Phase 3, randomized, open-label, multicentre	Adults n = 187	ALK-positive lung cancer	No prior therapy	Alectinib 600 mg twice per day or Crizotinib 250 mg twice per day	China, South Korea, Thailand	Alectinib: 4 (3) vs. Crizotinib: 12 (19)	Alectinib: 0 (0) vs. Crizotinib: 7 (11)
[47]	Retrospective, multicentre	Adults n = 1221	ALK-positive lung cancer	18% of patients had no prior therapy 81% had 1 or more prior lines of therapy	Alectinib 300 mg twice daily	Japan	93 (8)	14 (1)
[48]	Phase 1, open-label	Adults n = 246	ALK-positive lung cancer	19% of patients had no previous treatment regimen Other patients had 1–4 or more previous lines of treatment	Ceritinib 750 mg in dose escalation	Europe and North America and Asia-Pacific	NR	4 (2)
[49]	Phase 1, multicentre, open-label	Children n = 83	ALK-positive malignancies	55% of patients had 1–2 previous lines of therapy 25% of patients had 3 or more previous lines of therapy	Ceritinib dose escalation 500 mg/m^2^, 510 mg/m^2^	International	9 (11)	6 (7)
[50]	Phase 1, multicentre, open-label	Adults n = 20	ALK-positive lung cancer and inflammatory myofibroblastic tumours	80% had prior ALK therapy	Dose escalationCeritinib 300 mg, 450 mg, 600 mg, and 750 mg per day	Japan	4 (20)	0 (0)
[51]	Phase 3, multicentre, randomized, open-label	Adults n = 76	ALK-positive lung cancer	All patients had no prior therapy	Ceritinib 750 mg per day	Asia	8 (10)	2 (3)
[6]	Phase 3, randomized, controlled, open-label	Adults n = 115	ALK-positive lung cancer	All patients had crizotinib and 1 or 2 lines of chemotherapy	Ceritinib 750 mg per day	International	4 (3)	1 (2)
[18]	Phase 3, randomized, open-label, multicentre	Adults n = 189	ALK-positive lung cancer	No prior therapy	Ceritinib 750 mg per day	International	9 (5)	1 (1)
[10]	Phase 3, randomized, open-label, multicentre	Adults n = 11	ALK-positive lung cancer	All patients had 1 prior line of chemotherapy	Ceritinib 750 mg per day	Japan	1 (9)	0 (0)
[52]	Phase 3, multicentre, randomized, open-label	Adults n = 275	ALK-positive lung cancer	No prior therapy	Brigatinib 180 mg per day or Crizotinib 250 mg twice per day	International	Brigatinib: 3 (2) vs. Crizotinib: 13 (10)	Brigatinib: 3 (2) vs. Crizotinib: 11 (8)
[14]	Phase 3, randomized, open-label, multicentre	Adults n = 296	ALK-positive lung cancer	No prior therapy	Lorlatinib 100 mg per day or Crizotinib 250 mg twice per day	International	Lorlatinib: 10 (7) vs. Crizotinib: 21 (15)	Lorlatinib: 1 (1) vs. Crizotinib: 12 (8)
[53]	Phase 3, randomized, open-label, multicentre	Adults n = 25	ALK-positive lung cancer	No prior therapy	Lorlatinib 100 mg per day or Crizotinib 250 mg twice per day	Japan	Lorlatinib: 0 (0) vs. Crizotinib: 4 (18)	Lorlatinib: 0 (0) vs. Crizotinib: 3 (14)
[54]	Phase 1 and phase 2, single-arm, multicentre	Adults n = 193	NTRK tumours	73% of patients had prior chemotherapy	Entrectinib 600 mg/day	International	11 (6)	4 (2)
[55]	Phase 1 and phase 2, single-arm, multicentre	Adults n = 224	ROS1-positive lung cancer	63% of patients had prior systemic therapy	Entrectinib 600 mg/day	International	10 (4)	5 (2)
[56]	Phase 1/2, single-arm, multicentre	Children n = 43	ALK, ROS1, NTRK solid tumours or primary CNS tumours	77% of patients had prior chemotherapy	Dose escalationEntrectinib 250, 400, 550, 750 mg/m^2^	North America, Europe, North Korea	4 (9)	3 (7)
[57]	Phase 1, dose escalation	Adults n = 63	ALK- and ROS1- rearranged lung cancer	25% of patients had prior ALK-TKI treatment 48% of patients had chemotherapy	TQ-B3139 50–800 mg	China	8 (13)	0 (0)

Abbreviations: CNS: central nervous system; TKI: tyrosine kinase inhibitor.

**Table 2 cancers-15-04940-t002:** ALK inhibitors induced febrile neutropenia.

Ref.	Study Design	Population	Cancer Type	Treatment and Dose	Febrile Neutropenia (%)
[22]	Phase 1, open-label, multicentre	Children n = 79	ALK-positive solid tumours, lymphomas, CNS tumours	Dose escalation crizotinib 100, 130, 165, 215, 280, and 365 mg/m^2^ twice per day	10
[23]	Phase 2, multicentre, single-arm	Adults n = 1069	ALK-positive lung cancer	Crizotinib 250 mg twice per day	3
[25]	Phase 2, multicentre, single-agent, open-label	Adults n = 45	TFE3-rearranged alveolar soft part sarcoma	Crizotinib 250 mg twice per day	2
[44]	Phase 2, open-label, multicentre	Children and Adults n = 10	ALK-positive anaplastic large T-cell lymphoma	Alectinib 600 mg twice per day	10
[48]	Phase 1, open-label	Adults n = 255	ALK-positive lung cancer	Ceritinib 750 mg in dose escalation	<1
[6]	Phase 3, randomized, controlled, open-label	Adults n = 231	ALK-positive lung cancer	Ceritinib 750 mg per day	0

Abbreviations: CNS: central nervous system.

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
