# Peer review of "Anaplastic Lymphoma Kinase Inhibitor-Induced Neutropenia: A Systematic Review"

_cancers, 2023, doi:10.3390/cancers15204940_

Round 1
Reviewer 1 Report
This article nicely summarizes the existing reports of neutropenia induced by ALK inhibitors. Authors claimed that neutropenia was common after the performance of ALK inhibitors and appropriate dose restrictions may reduce side effects.
However, due to limitations in the number of existing literature, the value of this systematic review is limited. Readers can only know that corticosteroids can resume neutropenia, and the incidence of febrile neutropenia is low. The author can attempt to investigate the relationship between other side effects of ALK inhibitors and neutropenia so that make research results more substantial.
Besides, considering that neutrophils may play a promoting role in cancer development. Authors may analyze the relationship between neutrophilia and poor prognosis of lung cancer in the part of “Physiopathological hypotheses”.
Author Response
Reviewer #1:
This article nicely summarizes the existing reports of neutropenia induced by ALK inhibitors. Authors claimed that neutropenia was common after the performance of ALK inhibitors and appropriate dose restrictions may reduce side effects.
We thank the reviewer 1 for these kind comments
Comments :
However, due to limitations in the number of existing literature, the value of this systematic review is limited. Readers can only know that corticosteroids can resume neutropenia, and the incidence of febrile neutropenia is low. The author can attempt to investigate the relationship between other side effects of ALK inhibitors and neutropenia so that make research results more substantial.
We found no relationship in the literature between neutropenia and other side effects of these treatments.
Besides, considering that neutrophils may play a promoting role in cancer development. Authors may analyze the relationship between neutrophilia and poor prognosis of lung cancer in the part of “Physiopathological hypotheses”.
We added in the physiopathological hypotheses: “Inflammation is known to promote tumor development and angiogenesis, and to inhibit apoptosis [63]. Studies show that high neutrophil counts or a neutrophil-to-lymphocyte ratio ≥ 3 are associated with a poor prognosis [64]. Only one retrospective study of 36 patients showed a prognostic role for neutropenia induced by ALK inhibitors [58]”.

Reviewer 2 Report
The manuscript focuses on a systemicrevision of literature data about the adversae events observed in ALK positive NSCLC patients. In my opinion, minor integrations should be approaced to improve the readibility of the manuscript
- In the introduction section, please, could the authors overvuew clinical findings behind ALK inhibitors in NSCLC patiens? Please, could the authors add more details in this section?
-In the methodological section, please, could the authors evaluate the role of first and second line TKI anti ALK? In addition, could other molecular paramterers like as rearrfanged partner of expression levele may impact on the adversae events?
- In the table 1, please, could the authors explain why RET rearranged patients were included?
- In the discussion section, please, could the authors consider a clinical approach guided by this aspect?
Minor english revision
Author Response
Reviewer #2:
The manuscript focuses on a systemic revision of literature data about the adversae events observed in ALK positive NSCLC patients. In my opinion, minor integrations should be approaced to improve the readibility of the manuscript
We are deeply grateful for these comments.
In the introduction section, please, could the authors overview clinical findings behind ALK inhibitors in NSCLC patients? Please, could the authors add more details in this section? We added in the introduction section : “With a median follow-up time of 68.6 months with alectinib and 68.0 months with crizotinib, the median overall survival has still not been reached”.
In the methodological section, please, could the authors evaluate the role of first and second line TKI anti ALK? In addition, could other molecular parameters like as rearranged partner of expression level may impact on the adverse events? There is no difference in tolerance between the different ALK variants (D. Ross Camidge et al. J Thorac Oncol 2019). But neutropenia has not been studied in this situation.
In the table 1, please, could the authors explain why RET rearranged patients were included? We were interested in neutropenia induced by ALK inhibitors, whatever the treatment target. Alectinib has activity against RET in pre-clinical study but exerts limited activity in clinical study (Takeuchi et al. Transl Lung Cancer Res 2021).
In the discussion section, please, could the authors consider a clinical approach guided by this aspect? We added in the discussion session: “In view of the data in the literature, we propose the management of neutropenia induced by ALK inhibitors. In the event of neutropenia below 500mm3, we propose temporary interruption of treatment, followed by resumption with a reduction in dosage. In the event of recurrence, we suggest the introduction of corticosteroid therapy. This management is essential to maintain treatment, given its survival benefit for patients. There are no data in the literature on changing ALK inhibitor treatment. In our experience, in the event of recurrence of neutropenia despite discontinuation of treatment, changing the molecule does not eliminate this hematological toxicity”.
